# Isolation of *Mycobacterium avium* Subsp. *Paratuberculosis* in the Feces and Tissue of Small Ruminants Using a Non-Automated Liquid Culture Method

**DOI:** 10.3390/ani10010020

**Published:** 2019-12-20

**Authors:** Luigi De Grossi, Davide Santori, Antonino Barone, Silvia Abbruzzese, Matteo Ricchi, Gaetana Anita Marcario

**Affiliations:** 1Istituto Zooprofilattico Sperimentale del Lazio e della Toscana M. Aleandri, 00178 Roma, Italy; davide.santori@izslt.it (D.S.); antonino.barone@izslt.it (A.B.); silvia.abbruzzese@izslt.it (S.A.); anitamarcario@gmail.com (G.A.M.); 2Istituto Zooprofilattico Sperimentale della Lombardia e dell’Emilia Romagna B. Ubertini, 25124 Brescia, Italy; matteo.ricchi@izsler.it

**Keywords:** *Mycobacterium avium* subsp., *Paratuberculosis* (MAP), liquid media culture, sheep

## Abstract

**Simple Summary:**

In Italy, the field isolates of *Mycobacterium avium* subsp. *Paratuberculosis* in bovine and goats (type II or C) are frequently found while the isolation of MAP in sheep (type I or S) is very rare. The aim of this study was to detect field isolates of MAP in ovine and goats using a non-automated method with liquid media culture. For this purpose, five flocks were investigated by ELISA and some feces of positive animals were tested by quantitative polymerase chain reaction (qPCR). All direct PCR positive samples were tested using a non-automated culture method using liquid media. Positive samples were analyzed by typing specific-PCR in order to determine the strain type (I or II). Twenty-eight samples, feces and tissues, were cultivated and 12 of these were positive by qPCR. From 12 positive samples, six strains resulted type I (S), from two flocks and four sheep respectively. We have evaluated a rapid and inexpensive culture method for the recovering of MAP type I, which are mycobacteria extremely growth demanding.

**Abstract:**

Paratuberculosis is a chronic disease of ruminants caused by *Mycobacterium avium* subsp. *Paratuberculosis* (MAP). Since isolation of MAP type I (S) is rarely reported in Italy, our research was aimed at isolating, by an inexpensive liquid culture manual method, this type of MAP isolates. At first, we used an ELISA to point out to serologically positive samples from five flocks. Secondly, we used a fecal direct IS*900*-qPCR on the ELISA positive samples, in order to detect shedder animals. Feces from IS900-qPCR positive samples were inoculated in solid and liquid culture media. IS*900*-qPCR was further used to test the growth of MAP isolates in liquid medium, which were further confirmed by f57-qPCR and submitted to typing by specific PCR in order to identify the MAP type. Twenty-eight samples (24 fecal and four tissutal samples) were processed by culture methods, resulting in the isolation of six type I MAP field isolates. Notably, no isolates were recovered by solid media, underlining the utility of this liquid method. Few data about this type of MAP are currently available in Italy, and further analyses should be carried out in order to study the origin and epidemiology of type I strains circulating in Italy.

## 1. Introduction

Paratuberculosis (PTBC) or Johne’s disease is a chronic inflammation of the intestine caused by *Mycobacterium avium* subsp. *Paratuberculosis* (MAP). The disease mainly affects domestic and wild ruminants but can also affect other species [1]. In Europe, the spread appears to be growing in dairy cattle farms but also in sheep, goats, and wild animals [2,3,4,5,6,7].

PTBC leads to a significant decrease in production, weight loss and ultimately death [8,9,10]. It is estimated that for each animal presenting characteristic clinical symptoms, at the same time in the farm, between four to eight animals have a subclinical form and are therefore asymptomatic carriers [11]. The net cost of subclinical infection on our continent is estimated at around 250 euros per animal. The cost for sheep seems to be around 120 euros per year in dairy heads and 60 euros in meat animals. In addition to direct losses, there are also losses due to intercurrent diseases and infertility caused by PTBC [12,13]. Moreover, according to non-univocal pathogenic hypotheses, MAP could play a relevant role in the etiopathogenesis of Crohn’s disease [2]. In Italy MAP strains from bovine and goats (type II or C) are frequently isolated while the isolation of MAP from sheep is very rare, so the circulating field isolates and their characteristics are unknown. The aim of this study was to isolate MAP type I (S) field strains in ovine and goats using an inexpensive liquid media culture coupled with PCR for confirming the presence of MAP, since few information is currently available of MAP isolates circulating in ovine Italian flocks.

## 2. Materials and Methods

### 2.1. Study Design

Our research aimed at isolating MAP type I strains in sheep, since in our Country isolations of this type are extremely rare only showing three isolations. The main reason for this is because this type of MAP is extremely slow-growing and its isolation is very demanding, requiring until eight months for the first isolation.

In order to increase the chance of finding shedder animals and taking into account the relation between ELISA test and culture assays [14,15], we needed some positive and excretory samples. So, at first, we used an ELISA assay to point out serologically positive samples. Secondly, we used a fecal commercial IS900-qPCR to test if these ELISA positive animals were also shedding MAP. IS900-qPCR positive samples were then submitted to culture methods: one employing solid media and another employing liquid medium. For this second method, in order to check if MAP was growing, we periodically tested, by the same IS900-qPCR, portions of liquid medium. In order to increase the specificity of our liquid medium method, the same portions were also tested by f57-qPCR. Finally, in order to identify the type of MAP, the same samples were also submitted to typing by specific PCR [16,17]. (Figure 1)

Using data on the prevalence [18] of the ovine paratuberculosis in Latium region, it was possible to identify infected flocks and analyze them to search for MAP field isolates circulating in sheep and goat flocks [19,20]. Five flocks were investigated for paratuberculosis and one of these was considered to be free of paratuberculosis in previous research. Total analyzed are shown in Table 1.

### 2.2. Screening Samples Sera

Sera samples were analyzed by ELISA test (Id Screen Paratuberculosis Indirect Screening Test—ID-vet, Grabels, France) [21,22].

### 2.3. Culture Methods

Two g of feces were homogenized with 10 mL of diethyl pyrocarbonate (DEPC) water in 50 mL plastic tubes, placed on a mechanical agitator at 170 movements/min for 30 min. After 30 min of sedimentation, five mL of the supernatant were mixed with 12.5 mL of 1.5% 1-Hexadecylpyridinium Chloride (HPC) and 12.5 mL of BHI, vigorously vortexed and incubated at 37 ± 1 °C for 18–24 h. The tubes were centrifuged at 1200 *g* for 30 min at room temperature for the HPC solution to precipitate at low temperatures and then the supernatant was eliminated. The pellet was resuspended in one mL of the PANTA F (BD Diagnostics, Franklin Lakes, NJ, USA) antibiotic mixture and incubated at 37 ± 1 °C overnight. Two hundred microliters were inoculated into the following solid media: Herrold’s egg yolk medium, nalidixic acid, vancomycin, without mycobactin J (HEYM-ANV-SM); Herrold’s egg yolk medium, nalidixic acid, vancomycin, mycobactin J (HEYM-ANV-CM); and Herrold’s egg yolk medium, cloramphenicol (HEYM-CAF).

The samples subjected to double decontamination were also inoculated (200 µL) into Middlebrook 7H9 (BD Diagnostics) supplemented with egg yolk, mycobactin J (1 µg/mL) (ID-vet) [23] and a mixture of PANTA F antibiotics (2.5%) (BD Diagnostics) in 50 mL screw-capped tubes, leaving plenty of air and topping up with new medium periodically (every 40 days). This overcame the problems of evaporation occurring using smaller tubes which were suggested in a similar method [22] and left enough air to allow for the growth of MAP, which is an aerobic bacterium [24,25,26,27,28]. Periodically, time 0 and every 20–40 days, liquid cultures were tested by IS900-qPCR [29]. The positive samples to this test (17 feces, two ileocecal valves, two mesenteric lymph nodes) were further confirmed at the Italian National Reference Centre for paratuberculosis by f57-qPCR and then analyzed by type specific-PCR in order to determine the type I or II [19] (Table 2).

### 2.4. IS900-qPCR, f57-qPCR, type specific-PCR 

#### 2.4.1. DNA Extraction from Feces, Tissues, and Liquid Culture Media

Two g of feces were homogenized in 10 mL of DEPC water and placed on a mechanical agitator at 170 movements/min for 30 min. After 30 min of sedimentation, one mL was taken and centrifuged at 3000 *g* for five min. The pellet obtained was resuspended in 500 µL of DEPC water and submitted to DNA extraction by QIAamp DNA mini kit (Qiagen, Milan, Italy) according to the manufacturer’s instructions. For tissue extraction (ileocecal valve, mesenteric lymph nodes), two g were taken with a scalpel and homogenates in 10 mL of water in sterile filter bags. Five hundred µL of supernatant were subjected to extraction with QIAamp DNA mini kit (Qiagen) according to the manufacturer’s instructions. For each sample, DNA was eluted in 200 µL of elution buffer.

DNA extraction from non-automated liquid culture method required one mL of liquid medium placed in two mL conic safe-lock tubes and successive centrifugation at 13,200 *g*. Two hundred µL of supernatant were used to extraction by QIAamp DNA mini kit (Qiagen) according to the manufacturer’s instructions.

#### 2.4.2. IS900-qPCR

The PARATB REAL TIME kit (ADIAVET ™, Rochefort, Belgium) was used. The reactions were carried out in a Rotor-Gene Q (Qiagen) according to the manufacturer’s instructions [30].

#### 2.4.3. Other PCRs employed 

f57-qPCR [31] and type specific-PCR [19] were used according to previous studies [32,33,34,35].

#### 2.4.4. Pathological Examination

Two sheep were subjected to a full necropsy and macroscopic lesions were recorded. Tissue extracts were examined by hematoxylin and eosin (HE) and Ziehl-Neelsen (ZN) stains as previously reported [36,37,38,39,40,41,42].

## 3. Results

The farms were selected on the basis of the paratuberculosis prevalence in the territory identified in a previous study [16]. The five flocks tested included a flock of Saanen goats. Four were ELISA positive for paratuberculosis and one was confirmed to be negative and was used as control in order to check the absence of circulating MAP within the flock.

The results obtained for the fecal specimens are shown in Table 2. Considering all flocks, 86 out of 601 serum samples resulted positive for the ELISA test. During the blood collection, 417 fecal samples taken. The feces of 17 seropositive animals (with a high *S/P* value) and seven negatives (a total of 24 fecal samples), were used for direct IS900-qPCR and culture assays. Twelve of these were positive by direct fecal IS900-qPCR, but only eight were positive for IS900-qPCR carried out in liquid cultures, which were further confirmed by f57-qPCR. Finally, only six of these resulted as type I by type specific-PCR, probably because the amount of MAP DNA was not sufficient to allow the amplification by type specific-PCR. Interestingly, no type II strains were recovered, and, more importantly, no MAP isolates were recovered by solid media.

Two sheep died during the project and ileocecal valves and mesenteric lymph nodes were analyzed by direct IS900-qPCR. In the first subject both the ileocecal valve and the mesenteric lymph nodes, in addition to the feces, were positive by IS900-qPCR and by direct f57-qPCR. The liquid culture assay confirmed these results and type specific-PCR showed the presence of MAP type I after six weeks post inoculation. The anatomo-pathological investigation of the first subject revealed lesions compatible with paratuberculosis: enlarged mandibular and mesenteric lymph nodes, enteritis with cerebroid loops, and renal congestion; the histological examination showed severe diffuse granulomatous enteritis with presence of bacilli positive to Ziehl–Neelsen stain. In the second subject, tissues and feces were positive by direct IS900-qPCR, but were not confirmed by f57-qPCR and liquid culture assay. In this animal we observed: mild hyperplasia in the mesenteric lymph nodes, enteritis and renal congestion, catarrhal enteritis, characterized by the presence of inflammatory infiltrates in the *lamina propria*, consisting of lymphocytes, plasma cells, and eosinophils, with the presence of abscesses in the crypts and multiple areas of necrosis [39,40,41,42]. No bacilli positive to Ziehl–Neelsen stain were observed, but the ELISA *S/P* was still high (*S/P* 147%).

## 4. Discussion

The isolation of MAP type I strains is extremely rare in Italy. Multiple reasons for this have been identified, such as: type I are extremely slow-growing bacteria also requiring eight months for the first isolation; culture methods, for economic reasons, are not the first diagnostic choice when approaching paratuberculosis in sheep. Notably type I field isolates, but this seems to be true also for MAP Type II field isolates, prefer to grow in liquid media [31]. There are at least two systems in the market suitable for first isolation of MAP, but they are very expensive and this equipment can only be justified for large numbers of samples. In this study we evaluated a non-automated liquid medium culture method coupled with qPCR for the detection of MAP in feces and tissues of sheep and goats. Other studies have previously reported the superiority of automated liquid media over solid media for the culture of MAP in terms of sensitivity and earliness of results [43,44,45,46] and, more important, in our study, no MAP field isolates were recovered in solid media. The method herein employed has never been previously used for isolating MAP type I from sheep and goat samples, and can be easily adopted by many laboratories, with no sophisticated equipment [23]. The use of multiple methods, such as the ELISA test and direct IS900-qPCRs to detect shedder animals, greatly increased the possibility of isolating these demanding MAP type I strains by liquid culture medium [47]. Since a few MAP type I strains have been isolated in Italy, no epidemiological information is currently available about MAP type I circulating strains, so this method could be useful to fill this gap.

## 5. Conclusions

The liquid medium culture method for isolation of MAP type I field isolates is inexpensive, can be easily adopted by laboratories without access to sophisticated equipment, and can be used for processing many or few samples.

## Figures and Tables

**Figure 1 animals-10-00020-f001:**
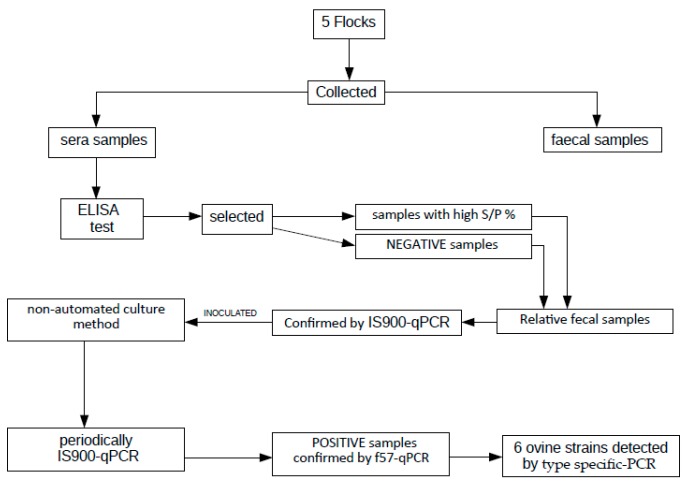
The methods employed in this study.

**Table 1 animals-10-00020-t001:** Samples collected in five flocks in Latium region.

	Sera	Faeces	Tissues
Sheep	496	339	4
Goats	105	78	0
Total	601	417	4

**Table 2 animals-10-00020-t002:** Results of non-automated liquid culture method.

	Number	Confirming IS900-qPCR	f57-qPCR	type specific-PCR
N Samples	28	12	8	6 ovine strains
N Animals	24	8	6	4 animals

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
