# Peer review of "Isolation of Mycobacterium avium Subsp. Paratuberculosis in the Feces and Tissue of Small Ruminants Using a Non-Automated Liquid Culture Method"

_animals, 2019, doi:10.3390/ani10010020_

Round 1
Reviewer 1 Report
The authors used a liquid culture followed by qPCR for detection of type I MAP in ovine fecal samples. The method is simply a combination of existing methods.
In Lines 25, 61 and 181-183, the authors claimed that the study aimed to “develop” a liquid culture method for isolation of type I MAP but that is at least an overstatement. Those sentences should be rephrased.
Line 50. Period is missing
Lines 50-52: References for Crohn’s disease should be cited.
Line 52: Delete “(MC)”. If the abbreviation is used in other places, change it to “CD”.
Figure 1: It would be good to add total number of samples in parenthesis.
Author Response
Dear Reviewer,
Thank you again for your consideration for our manuscript.
Q: The authors used a liquid culture followed by qPCR for detection of type I MAP in ovine fecal samples. The method is simply a combination of existing methods.
R: Indeed we have been greatly inspired by a previous paper which has been cited in the manuscript, but we have made some changes, otherwise, at least in our hands, the method did not work. The major change was the use of a greater volume of liquid medium and, consequently, the use of 50 ml screw tube in order to leave a sufficient volume of air to permit the growth of MAP and to allow the repeated testing by qPCR. Our modification to respect to the inspiring paper was already present in the M&M section of the original manuscript. Anyway, according to the following suggestion of the reviewer, we have changed the text (see below).
Q: In Lines 25, 61 and 181-183, the authors claimed that the study aimed to “develop” a liquid culture method for isolation of type I MAP but that is at least an overstatement. Those sentences should be rephrased.
R: The manuscript has been changed according to the comment of the reviewer and the concept related to the word “develop” has been changed with the concept related to the word “used”. We hope this new revision match the request of the reviewer.
Q: Line 50. Period is missing
R: The entire sentence was removed as considered redundant in the new version of the manuscript.
Q: Lines 50-52: References for Crohn’s disease should be cited.
R: Reference number 2 was added in the manuscript.
Q:Line 52: Delete “(MC)”. If the abbreviation is used in other places, change it to “CD”.
R: The abbreviation was deleted since it was not more used in the text
Q: Figure 1: It would be good to add total number of samples in parenthesis.
R: We respectfully disagree with the reviewer because the number of samples analysed is already reported in tables 1 and 2. For this reason, we believe adding these information also into the figure, does not help the readability of the paper and instead ballast the figure. So we would prefer to maintain this version of the figure. We hope both the reviewer and the editor agree with our decision.
Reviewer 2 Report
English needs to be improved, the text in some places is not very smooth with many repeated words.
Line 56: please change "no information" with "few information".
Line 89: Missing information on the ELISA assay.
Line 181: please change "developed" with "evaluated".
Discussions and conclusions need to be improved.
I think that the paper can be accepted as Short Communication after revision.
Author Response
Dear Reviewer,
Thank you again for your consideration for our manuscript.
Q. English needs to be improved, the text in some places is not very smooth with many repeated words.
R: The text has been further revised in order to delete or simplify the text. Some sentences have been removed since we found they were ballasting the text. In particular we have removed the following sentences form the second submission of our manuscript:
Sentence at line 50;
Sentence at lines 186-188;
Sentence at line 195.
Moreover, we have carefully checked the text in order to try to remove, when possible, the repeated words. We don’t know if we were able to succeed, but we hope this new version is able to satisfy the reviewer.
Q: Line 56: please change "no information" with "few information".
R: We agree, it has been done as requested.
Q: Line 89: Missing information on the ELISA assay.
R: Further information about the company have been added in the text.
Q: Line 181: please change "developed" with "evaluated".
R: Also reviewer 1 raised a similar comment, we agree thanking both reviewer to have raised this point and course the suggestion has been accepted.
Q: Discussions and conclusions need to be improved.
R: We have tried to improve both sections by removing some sentence and repeated words which ballasted the paper. Moreover, we have also tried to be more specific in our sentences considering also the comments of reviewer 1. We hope this new version is able to satisfy the reviewer.
Q: I think that the paper can be accepted as Short Communication after revision.
R. Indeed, we feel the reviewer is right suggesting this manuscript is better suited as short communication, but since it should be included in a special number of Animals, we don’t know if this arrangement is fine with the editor
This manuscript is a resubmission of an earlier submission. The following is a list of the peer review reports and author responses from that submission.
Round 1
Reviewer 1 Report
Author analyzed sera samples from 5 flocks containing 601 animals (sheep and goats) to screen seropositive animals for Mycobacterium avium Subsp. Paratuberculosis (MAP) using indirect ELISA. Moreover, author analyzed faces from positive supershedder animals. IS900-qPCR was used for detection of MAP. Positive infected samples were further used for solid and liquid bacterial culture. Again IS900-qPCR and f57-qPCR was used to confirm the presence of MAP in bacterial culture. Finally, DMC-PCR was used to identify strains type (I or II).
Author has provided good rational for the current study. However, there are many minor issues that need to be address before the publication.
I have listed few of the issues author should consider during revision.
First of all, manuscript need to be sent for English correction from native speaker. It is hard for the reader to understand what message author want to convey.
Scientific writing need to be strongly improved. This is a scientific journal and authors should pay attention to the good writing.
Line 17-18: “Among them…and four sheep”. The meaning of the sentence is not clear. Please rewrite the sentence.
In abstract, author mentioned three types of PCRs (IS900-qPCR, f57-qPCR and DMC-PCR). Please specify fill forms of each PCRs and which PCR is used for which samples. Form the current description, it is hard to understand how the experiment was designed and which samples were used for analysis.
Please provide introduction on IS900-qPCR, f57-qPCR and DMC-PCR and why they are used for specific samples. Have three qPCRs used before by another researcher? Kindly provide a brief paragraph in the introduction section.
Four tissue samples were used to extract DNA for detection of MAP Table 1. Author mentioned three tissues (intestine, ileocecal valve, mesenteric lymph node) were taken from sheep (line 96). Please provide sufficient information which 4 samples were used for current study.
Correct any typos, e.g. Line 70, 117, 171 etc.
Line 129: Kindly provide image of Ziehl-Neelsen staining.
Line 72: What do you mean by 1.5% HPC? Is it 1.5 % hydroxypropylcellulose? Kindly provide full forms of all abbreviations used in the manuscript.
There is inconsistency in writing numbers. Sometime author write number in words and sometime in number. E.g. Line 100 and 101: “Two hundred µL” and “1 mL”.
Line 100: What is the meaning of 2,5 mL? Does author want to say 2.5 mL? Please use universally accepted system for writing number with decimal points.
Line 101: …centrifugation at 13200 RPM/min”. Which centrifuge machine was used? RPM doesn’t provide accurate information, kindly provide centrifugation values in “g”.
Kindly improve the quality of the discussion section. Roughly we would write each paragraph of the discussion considering three points. Point 1: Report what you observed. Point 2: Report what other researchers have reported. Point 3: Write your conclusion.
E.g. Gene X is regulated in Dogs (point 1). Earlier, gene X was shown to be regulated in cat sheep and goats but not in tiger, lion and deer (ref) (point 2). We conclude that gene X is regulated in domestic animal but may not be regulated in wild animals (point 3).
Author Response
Dear reviewer, thank you for your suggestions and observations.
Point by point list of answers to questions raised by the reviewer, our answers are shown in red:
Author analyzed sera samples from 5 flocks containing 601 animals (sheep and goats) to screen seropositive animals for Mycobacterium avium Subsp. Paratuberculosis (MAP) using indirect ELISA. Moreover, author analyzed faces from positive supershedder animals. IS900-qPCR was used for detection of MAP. Positive infected samples were further used for solid and liquid bacterial culture. Again IS900-qPCR and f57-qPCR was used to confirm the presence of MAP in bacterial culture. Finally, DMC-PCR was used to identify strains type (I or II).
Author has provided good rational for the current study. However, there are many minor issues that need to be address before the publication.
I have listed few of the issues author should consider during revision.
First of all, manuscript need to be sent for English correction from native speaker. It is hard for the reader to understand what message author want to convey.
Scientific writing need to be strongly improved. This is a scientific journal and authors should pay attention to the good writing.
A. We thank the reviewer for his comments and precious suggestions. Indeed the new version the manuscript has been extensively revised with the help of a native English speaker. Moreover, in many parts, the manuscript has been simplified in the attempt to increase the readability
Line 17-18: “Among them…and four sheep”. The meaning of the sentence is not clear. Please rewrite the sentence.
A. We changed sentence in line 17-18 .
In abstract, author mentioned three types of PCRs (IS900-qPCR, f57-qPCR and DMC-PCR). Please specify fill forms of each PCRs and which PCR is used for which samples. Form the current description, it is hard to understand how the experiment was designed and which samples were used for analysis.
Please provide introduction on IS900-qPCR, f57-qPCR and DMC-PCR and why they are used for specific samples. Have three qPCRs used before by another researcher? Kindly provide a brief paragraph in the introduction section.
A. We provided to insert a study design chapter in materials and methods section in order to clarify the use of three PCRs and how the experiment was designed.
Four tissue samples were used to extract DNA for detection of MAP Table 1. Author mentioned three tissues (intestine, ileocecal valve, mesenteric lymph node) were taken from sheep (line 96). Please provide sufficient information which 4 samples were used for current study.
A. Four tissue were 2 ileocecal valve and 2 mesenteric lymph node were added in the appropriate section of the manuscript . The word “intestine” was a mistake and was deleted.
Correct any typos, e.g. Line 70, 117, 171 etc.
Line 129: Kindly provide image of Ziehl-Neelsen staining.
Line 72: What do you mean by 1.5% HPC? Is it 1.5 % hydroxypropylcellulose? Kindly provide full forms of all abbreviations used in the manuscript.
There is inconsistency in writing numbers. Sometime author write number in words and sometime in number. E.g. Line 100 and 101: “Two hundred µL” and “1 mL”.
Line 100: What is the meaning of 2,5 mL? Does author want to say 2.5 mL? Please use universally accepted system for writing number with decimal points.
Line 101: …centrifugation at 13200 RPM/min”. Which centrifuge machine was used? RPM doesn’t provide accurate information, kindly provide centrifugation values in “g”.
Kindly improve the quality of the discussion section. Roughly we would write each paragraph of the discussion considering three points. Point 1: Report what you observed. Point 2: Report what other researchers have reported. Point 3: Write your conclusion.
E.g. Gene X is regulated in Dogs (point 1). Earlier, gene X was shown to be regulated in cat sheep and goats but not in tiger, lion and deer (ref) (point 2). We conclude that gene X is regulated in domestic animal but may not be regulated in wild animals (point 3).
A. All suggestions have been accepted but unfortunately we it was not possible to recover any pictures of Ziehl-Neelsen stain during the analysis of this sample because our labs are not provided with specific microscopic camera. Anyway, we are sure about the nature of MAP isolates because of results obtained with the molecular assays employed. We have also tried to improve the discussion section taking in mind the suggestion raised by the reviewer, we hope the new version is more readable and more scientific sounding.
Reviewer 2 Report
The authors tested sera, feces and tissues of sheep and goats for paratuberculosis infection by using ELISA and PCR tests. Although a good number of samples tested, the result was poorly presented. In short, the manuscript is not acceptable for a publication in its current form. There are numerous minor comments but below I list only major concerns.
The title does not represent what the finding was.
The Simple Summary should be re-written. The whole manuscript needs to be extensively edited by a native English speaker (science editor).
It is not clear what was the hypothesis and what the author aimed to find out, and what the contribution of this study to scientific community is.
It is not clear how many of the collected samples were tested by each of the method. I would recommend combining Tables 1 and 2 and place it in the result section.
PCR tests were conducted only on ELISA positive samples. I understand this is a measure used for PTBC control but is not so meaningful in terms of epidemiological study (as there should be many PCR positives in ELISA negatives).
The histological analysis procedure should be described in detail in the method section.
The conclusion is not supported by the data presented.
Author Response
Dear reviewer, thank you for your suggestions and observations.
Point by point list of answers to questions raised by the reviewer, our answers are shown in red:
The authors tested sera, feces and tissues of sheep and goats for paratuberculosis infection by using ELISA and PCR tests. Although a good number of samples tested, the result was poorly presented. In short, the manuscript is not acceptable for a publication in its current form. There are numerous minor comments but below I list only major concerns.
The title does not represent what the finding was.
A. Our research tried to use a manual method with liquid culture media as a new method to isolate MAP in sheep since the isolation of MAP strains from this species is rare in our country. In the manuscript we have also discussed these specific points. Anyway, we have changed the title in order to make it more suited for the content of manuscript.
The Simple Summary should be re-written. The whole manuscript needs to be extensively edited by a native English speaker (science editor).
A. The entire manuscript, including the simple summary, has been deeply revised with the help of a native English speaker and with the help of expert colleagues in order to ameliorate the readability and scientific sounding of the manuscript. We hope the new version can meet the comments and suggestions raised by the reviewer.
It is not clear what was the hypothesis and what the author aimed to find out, and what the contribution of this study to scientific community is.
It is not clear how many of the collected samples were tested by each of the method. I would recommend combining Tables 1 and 2 and place it in the result section.
A. We have inserted a new chapter in materials and methods section called “study design” in order to clarify the use of three PCRs and how the experiment was designed. In addition we inserted a flow chart of the experimental study to further clarify all these fundamental aspects.
PCR tests were conducted only on ELISA positive samples. I understand this is a measure used for PTBC control but is not so meaningful in terms of epidemiological study (as there should be many PCR positives in ELISA negatives).
A. The goal of our paper was not reporting epidemiological information about the paratuberculosis in this area but instead reporting information about the use of a manual method based on liquid culture medium aimed at isolating MAP strains type I, which are very demanding and fastidious MAP strains. We have used previous epidemiological data (serological data) about the paratuberculosis in this area, which has already been implemented in Lazio and Tuscany, according to previous publications about the diffusion and prevalences of paratuberculosis in Europe, more than 50% of cattle and sheep farms are infected. However, the intra flock prevalence is low, around 3-5%, so, in order to recover about 20-30 positive samples from shedding animals, we should have tested by cultural method around 600 samples, taking also in account that MAP elimination through faeces is intermittent and not continuous. In other words, we have used ELISA test and direct qPCR as prompt to select the samples from shedding animals
The histological analysis procedure should be described in detail in the method section.
A. This was done in the material and methods section.
The conclusion is not supported by the data presented.
A. Conclusion have been simplified according to the content manuscript, we hope this new version can meet the suggestion of the reviewer
Reviewer 3 Report
Dear Authors,
our paper "Mycobacterium avium subsp. Paratuberculosis in feces and tissue of small ruminants using a non-automated liquide culture system method" can be accepted for publication in "Animals" after major revision.
The level of scientific writing in English is insufficient to justify publication. The manuscript needs to be redacted by a native speaker of English with experience in scientific writing. There are many paragraphs that sound like a literal translation of a text previously written in Italian.
The paper is completely lacking in statistical analysis and data analysis.
Some equipment is named with the brand name (Falcon, Eppendorf), I think it is appropriate to use a generic name (es: 1.5 ml microtube).
The complete company name must be provided (address, code, etc.).
Lines from 60 to 65: The authors state that ELISA assay was used to find seropositive animals, have they considered the possibility of false-positives and false-negative results?
Furthermore, only the positive fecal samples for IS900-PCR were analyzed by cultural method, have considered the authors that the use of a test in a high prevalence population/samples increases the positive predictive value of the tests?
How did the authors establish the negativity of control?
The results are confusedly described (especially lines from 111 to 120), and the tables don’t clarify the text.
The acronyms must be explained.
In the discussion the authors talking about “analytical sensitivity” (line 149). Analytical sensitivity is the assay's ability to detect very low concentrations of a given substance in a biological specimen and it is also referred to as the limit of detection (LoD). How did the authors establish the analytical sensitivity value?
How did the authors compare the different tests, and how did they calculate sensitivity and specificity?
Finally, discussion and conclusion must be implemented.
Author Response
Dear reviewer, thank you for your suggestions and observations.
Point by point list of answers to questions raised by the reviewer, our answers are shown in red:
your paper "Mycobacterium avium subsp. Paratuberculosis in feces and tissue of small ruminants using a non-automated liquide culture system method" can be accepted for publication in "Animals" after major revision.
The level of scientific writing in English is insufficient to justify publication. The manuscript needs to be redacted by a native speaker of English with experience in scientific writing. There are many paragraphs that sound like a literal translation of a text previously written in Italian.
A. The entire manuscript, has been deeply revised with the help of a native English speaker and with the help of expert colleagues in order to ameliorate the readability and scientific sounding of the manuscript. We hope the new version can meet the comments and suggestions raised by the reviewer.
The paper is completely lacking in statistical analysis and data analysis.
A. The goal of our paper was not reporting epidemiological information about the paratuberculosis in this area, which has already been implemented in Lazio and Tuscany, but instead reporting information about the use of a manual method based on liquid culture medium aimed at isolating MAP strains type I, which are very demanding and fastidious MAP.
Some equipment is named with the brand name (Falcon, Eppendorf), I think it is appropriate to use a generic name (es: 1.5 ml microtube).
A. Done as suggested.
The complete company name must be provided (address, code, etc.).
A. Done as suggested.
Lines from 60 to 65: The authors state that ELISA assay was used to find seropositive animals, have they considered the possibility of false-positives and false-negative results?
Furthermore, only the positive fecal samples for IS900-PCR were analyzed by cultural method, have considered the authors that the use of a test in a high prevalence population/samples increases the positive predictive value of the tests?
How did the authors establish the negativity of control?
A. The ELISA, as well as direct Is900-qPCR, were used as prompt to select the samples from shedding animals and no comparison among the different method was carried out in the present study. Anyway, we are well aware ELISA test has low but variable sensitivity and high specificity (99%), so the risk to have false positive samples is rare, especially when it is confirmed by PCR tests. Instead, all paratuberculosis assay have a low sensitivity. But our intention was just to monitor the negative samples. That’s why we used a flock whit a negative status already known .
The results are confusedly described (especially lines from 111 to 120), and the tables don’t clarify the text.
A. Results section has been revised in order to make it more readable, we hope now it’s clear.
The acronyms must be explained.
A. Acronyms were written in full
In the discussion the authors talking about “analytical sensitivity” (line 149). Analytical sensitivity is the assay's ability to detect very low concentrations of a given substance in a biological specimen and it is also referred to as the limit of detection (LoD). How did the authors establish the analytical sensitivity value?
How did the authors compare the different tests, and how did they calculate sensitivity and specificity?
Finally, discussion and conclusion must be implemented.
A. The entire manuscript, has been deeply revised with the help of a native English speaker and with the help of expert colleagues in order to ameliorate the readability and scientific sounding of the manuscript. We hope the new version can meet the comments and suggestions raised by the reviewer.
Reviewer 4 Report
Dear Editor,
I revised the manuscript entitled “Detection of Mycobacterium avium subsp. Paratuberculosis in faeces and tissue of small ruminants using a non - automated liquide culture system method ” by De Grossi, D. Santori, A. Barone, S. Abbruzzese, M. Ricchi, G.A. Marcario.
The manuscript is presented as an original paper, the study results are interesting and adequate to your Journal and its topic is of interest for researchers working in the field.
Therefore, considering the relevance of the topic, and the originality of the results, I suggest it for publication on “Animals” pending minor revision; suggesting the following changes to the authors in order to improve the manuscript.
Specific comments:
Taxonomy of microorganisms: Genus in uppercase and species in lowercase, both in italics. To be corrected also in the References. Mycobacterium avium subsp. paratuberculosis.
Standardize the words throughout the text: faeces (feces), culture (cultural), type I ( Type I) analysed (analyzed), IS900 qPCR, F57 qPCR, ( IS900 qPCR, f57 qPCR).
Title: I would like to suggest: Detection of Mycobacterium avium subsp. paratuberculosis in faeces and tissue of small ruminants using a non - automated liquid culture method.
Simple Summary:
Line 13 I would suggest that the method used be standardized throughout the text like: non-automated method with liquid media culture.
Line 14-15 Please change into: All PCR positive samples were tested with a non-automated culture method.
Abstract
Line 28 Please change “filed” into “field”
Line 30 I would avoid repeating “in Italy”.
Keywords: Mycobacterium avium subsp. paratuberculosis (MAP);
Introduction
Line 51 Please change “using a cultural liquid manual method” into “with a non-automated method using a liquid media culture”
Table 1 Please change “Collected” into “Total”
Line 61 some faeces (How many?) of positive supershedder animals by…
2.2 Cultural methods
Line 72 Please specify the manufacturing company of the following media: HPC and BHI.
Lines 77,78,79: Please specify the manufacturing company of the following solid media: Herrold's Egg Yolk Medium, Nalidixic Acid, Vancomycin, without Mycobactin J (HEYM-ANV-SM), Herrold’s Egg Yolk Medium, Nalidixic Acid, Vancomycin, Mycobactin J (HEYM-ANV-CM), Herrold’s Egg Yolk Medium, Cloramphenicol (HEYM-CAF).
Line 86 The positive samples
Line 87 Please specify which Reference Center National (CRN) and where is located.
Line 89 Table 2 Positive samples and animals at culture method, IS900 qPCR, F57 qPCR, DMC PCR.
Line 89 Table n.2: the origin of the 28 samples is not very clear. The authors could better explain this result in the text.
Line 94 centrifuged at 3000 g (I council to standardize in RPM / min)
Line 100 Please change “1 mL” into “ 1 ml”
Line 100 Please change “eppendorf” into “conic safe-lock tubes” (Eppendorf)
Line 101 13,200 RPM/min
Line 107 manufacturer's
3.Results
Line 112 “Sahanen” into “Saanen”
Line 114 Table 1 (Table 2) or make a new clearer Results Table
Line 116 416 (417) faecal samples
Lines 118-20 Finally, only six of these resulted as type I (S) by DMC- PCR, samples were from two flocks and four sheep (as stated in the Abstract) while the rest of the samples resulted negative probably because the amount of MAP DNA was not sufficient to allow the amplification by DMC-PCR.
Line 129 stainning (staining)
Line 139 fot for
Line 139 ATCC 1368( Type C) strain, (Sample check to be inserted in Materials and methods)
Line 142 MAP isolates (strains)
Line 146 ( Round bracket to be deleted.
Line 154 “Middlebrook 7H9” or “M7H9”
Line 158 Pozzato [17] which in turn has changed that what was effective even with a few cultivated samples.
Line 160 sheep MAP strains
Line 160 .. Delete a point.
Line 161 Add (the) research
Line 161 (to discover infected flocks). It is pleonastic.
Line 162 the difficulty of MAP cultivation is mainly due to the long time required for the growing
Line 163 the liquid culture
Line 164 making difficult any typing test with PCR.
Line 164 Please change “modified” into “replaced”
Line 169 resulted positive
Line 169 liquid cultural culture
Line 170 Disconnect: certainly influenced
Line 170 demonstrated
Line 174 MAP type I
Line 174 sheep MAP strains
Line 177 Middlebrook7H9 o M7H9
References In my opinion, Volume and pp must be eliminated in all the references.
Line 185: 2001,
Line 206 alpacas (Lama pacos)
Line 232-233 Mycobacterium johnei, M. johnei
Line 245 In my opinion, Issues must be eliminated.
Best regards
Author Response
Dear reviewer, thank you for your suggestions and observations.
Point by point list of answers to questions raised by the reviewer, our answers are shown in red:
Specific comments:
Taxonomy of microorganisms: Genus in uppercase and species in lowercase, both in italics. To be corrected also in the References. Mycobacterium avium subsp. paratuberculosis.
Standardize the words throughout the text: faeces (feces), culture (cultural), type I ( Type I) analysed (analyzed), IS900 qPCR, F57 qPCR, ( IS900 qPCR, f57 qPCR).
Title: I would like to suggest: Detection of Mycobacterium avium subsp. paratuberculosis in faeces and tissue of small ruminants using a non - automated liquid culture method.
Simple Summary:
Line 13 I would suggest that the method used be standardized throughout the text like: non-automated method with liquid media culture.
Line 14-15 Please change into: All PCR positive samples were tested with a non-automated culture method.
Abstract
Line 28 Please change “filed” into “field”
Line 30 I would avoid repeating “in Italy”.
Keywords: Mycobacterium avium subsp. paratuberculosis (MAP);
Introduction
Line 51 Please change “using a cultural liquid manual method” into “with a non-automated method using a liquid media culture”
Table 1 Please change “Collected” into “Total”
Line 61 some faeces (How many?) of positive supershedder animals by…
2.2 Cultural methods
Line 72 Please specify the manufacturing company of the following media: HPC and BHI.
Lines 77,78,79: Please specify the manufacturing company of the following solid media: Herrold's Egg Yolk Medium, Nalidixic Acid, Vancomycin, without Mycobactin J (HEYM-ANV-SM), Herrold’s Egg Yolk Medium, Nalidixic Acid, Vancomycin, Mycobactin J (HEYM-ANV-CM), Herrold’s Egg Yolk Medium, Cloramphenicol (HEYM-CAF).
Line 86 The positive samples
Line 87 Please specify which Reference Center National (CRN) and where is located.
Line 89 Table 2 Positive samples and animals at culture method, IS900 qPCR, F57 qPCR, DMC PCR.
Line 89 Table n.2: the origin of the 28 samples is not very clear. The authors could better explain this result in the text.
Line 94 centrifuged at 3000 g (I council to standardize in RPM / min)
Line 100 Please change “1 mL” into “ 1 ml”
Line 100 Please change “eppendorf” into “conic safe-lock tubes” (Eppendorf)
Line 101 13,200 RPM/min
Line 107 manufacturer's
3.Results
Line 112 “Sahanen” into “Saanen”
Line 114 Table 1 (Table 2) or make a new clearer Results Table
Line 116 416 (417) faecal samples
Lines 118-20 Finally, only six of these resulted as type I (S) by DMC- PCR, samples were from two flocks and four sheep (as stated in the Abstract) while the rest of the samples resulted negative probably because the amount of MAP DNA was not sufficient to allow the amplification by DMC-PCR.
Line 129 stainning (staining)
Line 139 fot for
Line 139 ATCC 1368( Type C) strain, (Sample check to be inserted in Materials and methods)
Line 142 MAP isolates (strains)
Line 146 ( Round bracket to be deleted.
Line 154 “Middlebrook 7H9” or “M7H9”
Line 158 Pozzato [17] which in turn has changed that what was effective even with a few cultivated samples.
Line 160 sheep MAP strains
Line 160 .. Delete a point.
Line 161 Add (the) research
Line 161 (to discover infected flocks). It is pleonastic.
Line 162 the difficulty of MAP cultivation is mainly due to the long time required for the growing
Line 163 the liquid culture
Line 164 making difficult any typing test with PCR.
Line 164 Please change “modified” into “replaced”
Line 169 resulted positive
Line 169 liquid cultural culture
Line 170 Disconnect: certainly influenced
Line 170 demonstrated
Line 174 MAP type I
Line 174 sheep MAP strains
Line 177 Middlebrook7H9 o M7H9
References In my opinion, Volume and pp must be eliminated in all the references.
Line 185: 2001,
Line 206 alpacas (Lama pacos)
Line 232-233 Mycobacterium johnei, M. johnei
Line 245 In my opinion, Issues must be eliminated.
Best regards
A. Since some part of the manuscript have been so deeply changed, it was not possible to accept all suggestions raised by the reviewer, but those in those parts which remained unchanged, all suggestions have been accepted.